# Graph Theory-Based Electroencephalographic Connectivity via Phase-Locking Value and Its Association with Ketogenic Diet Responsiveness in Patients with Focal Onset Seizures

**DOI:** 10.3390/nu14214457

**Published:** 2022-10-23

**Authors:** Tzu-Yun Hsieh, Pi-Lien Hung, Ting-Yu Su, Syu-Jyun Peng

**Affiliations:** 1Division of Pediatric Neurology, Department of Pediatrics, Kaohsiung Chang Gung Memorial Hospital, Chang Gung University College of Medicine, Kaohsiung 83301, Taiwan; 2Professional Master Program in Artificial Intelligence in Medicine, College of Medicine, Taipei Medical University, Taipei 10675, Taiwan

**Keywords:** ketogenic diet (KD), focal-onset epilepsy, child behavior checklist (CBCL), EEG functional connectivity, phase-locking value (PVL)

## Abstract

Ketogenic diets (KDs) are a promising alternative therapy for pediatric refractory epilepsy. Several predictors of KD responsiveness have been identified, including biochemical parameters, seizure types, and electroencephalography (EEG) examinations. We hypothesized that graph theory-based EEG functional connectivity could explain KD responses in patients presenting focal onset seizure (FOS). A total of 17 patients aged 0–30 years old with focal onset seizures (FOS) were recruited as a study group between January 2015 and July 2021. Twenty age-matched children presenting headache with no intracranial complications nor other medical issues were enrolled as a control group. Data were obtained at baseline and at 12 months after initiating KD therapy (KDT) using the child behavior checklist (CBCL) and brain functional connectivity parameters based on phase-locking value from 19 scalp EEG signals, including nodal strength, global efficiency, clustering coefficient, and betweenness centrality. Compared with age-matched controls, patients presenting FOS with right or bilateral EEG lateralization presented higher baseline functional connectivity, including parameters such as global efficiency, mean cluster coefficient and mean nodal strength in the delta and beta frequency bands. In patients presenting FOS with right or bilateral EEG lateralization, the global efficiency of functional connectivity parameters in the delta and theta frequency bands was significantly lower at 12 months after KDT treatment than before KDT. Those patients also presented a significantly lower mean clustering coefficient and mean nodal strength in the theta frequency band at 12 months after KDT treatment. Changes in brain functional connectivity were positively correlated with social problems, attention, and behavioral scores based on CBCL assessments completed by parents. This study provides evidence that KDT might be beneficial in the treatment of patients with FOS. Graph theoretic analysis revealed that the observed effects were related to decreased functional connectivity, particularly in terms of global efficiency. Our findings related to brain connectivity revealed lateralization to the right (non-dominant) hemisphere; however, we were unable to define the underlying mechanism. Our data revealed that in addition to altered brain connectivity, KDT improved the patient’s behavior and emotional state.

## 1. Introduction

Ketogenic diets (KDs) promote high-fat, low-carbohydrate, and adequate protein level regimen while preserving a normal amount of calories intake. KDs have been widely acknowledged as a non-pharmacologic therapy for intractable childhood epilepsy. Meta-analysis has revealed that 52% of patients who followed a KD experienced a ≥90% reduction in the incidence of seizures, while 24% experienced seizure freedom [1]. The only predictable responses to KD therapy (KDT) are glucose transporter type 1 deficiency syndrome, pyruvate dehydrogenase deficiency, and specific mitochondrial disorders [2]. The mechanisms underlying the antiseizure effects of KDs have yet to be elucidated; however, it has been posited the elevated serum levels of ketone bodies and/or polyunsaturated fatty acids (PUFAs) play a role. Elevated ketone level has been shown to (1) increase the expression of inhibitory neurotransmitters, (2) activate potassium channels leading to neuronal hyperpolarization, and (3) increase energy production in the nervous system. Any of these effects could conceivably increase the seizure threshold. Elevated PUFA levels have been shown to activate peroxisome proliferator-activated receptors (PPARs) which prevent neuronal hyperexcitability, inhibition of voltage-gated sodium and calcium channels, activate potassium channels leading to neuron hyperpolarization, activate sodium/potassium/adenosine triphosphatase (Na+/K/ATPase), and upregulate and activate of uncoupling proteins (UCPs) leading to decrease in reactive oxygen species (ROS) production and oxidative stress. These effects could conceivably decrease the incidence of seizures [3].

Several factors have been used to predict the effectiveness of KD, including age, gender, intellectual status, biochemical parameters [4,5], and EEG parameters [6]. KD is an established therapy aimed at moderating seizures in patients with pharmaco-resistant epilepsy; however, its effects on cerebral neurophysiology have yet to be fully elucidated. Previous research has shown a significant reduction in the number of interictal epileptiform discharges (IEDs) after 3 months of KDT [7]. The presence of IED in the temporal region is associated with a poor response to KDT [8]. Among 18 children undergoing 24-h EEG monitoring, researchers reported a reduction in the median spike frequency after KDT, particularly during sleep [7]. Children who presented an improvement of more than 10% in IED frequency after KDT 1 month were 6 times more likely to be responsive to KDT [9]. One study reported a 30% reduction in the IED index in sleep EEG data after 6 weeks of KDT was associated with good KDT effectiveness [10]. In our previous work, we proposed graph theory-based EEG functional connectivity as a feasible approach to monitoring the effectiveness of KDT in children with epilepsy [11]. However, it should be noted that researchers have yet to reach a consensus as to the benefits of KDT.

Children with newly diagnosed epilepsy are significantly more likely than healthy children to present psychiatric comorbidities, including anxiety, depressive disorders, attention deficit hyperactivity/impulsivity disorder, oppositional-provocative disorder, and tic disorders [12]. Researchers have reported that KD can improve cognition and attention, while alleviating behavioral problems in children with epilepsy [13,14,15]. Animal and human studies have both provided evidence that KD is beneficial in regulating mental, emotional, and behavioral problems [16]. Nonetheless, the mechanisms underlying these benefits are poorly understood.

Brain connectivity is a high complex process involving inhibitory and excitatory signals at local and global levels [17]. Brain connectivity can be discussed from the perspective of connectivity via anatomical tracts or functional associations [18]. Graph theory methods have been widely used to analyze the brain connectivity and quantify the macroscopic structural and functional properties of brain networks [19]. Functional segregation refers to the ability to interconnect local brain regions. Segregation is typically measured by the number of triangles in a network and the fraction of triangles around an individual node is known as the clustering coefficient. Functional integration refers to the ability to rapidly combine information from distributed brain regions. The extent of integration indicates the ease with which brain regions communicate, which is typically measures in terms of global efficiency [18]. The belief that seizures are the result of an imbalance between inhibitory and excitatory signals in brain connectivity has prompted many researchers to explore the role of functional connectivity in seizures [20,21,22,23]. One study using EEG to study fronto-temporal focal epilepsy reported increased functional connectivity (local segregation as well as global integration) in the affected hemisphere [22].

In the current study, we employed graph theoretic analysis based on phase-locking value (PVL) to study of EEG functional connectivity with the aim of elucidating the effects of KD on brain network connectivity in patients with FOS. We also investigated the means by which alterations in brain networks manifest as emotional and behavioral changes.

## 2. Materials and Methods

### 2.1. Participants

This retrospective study included outpatients with FOS who underwent KD between January 2015 and July 2021.We also recruited a control group of age-matched children presenting headache with no intracranial complications nor other medical issues. This study was performed under the supervision of the Institutional Review Board at Chang Gung Memorial Hospital (IRB number: 202101494B0). Informed consent was obtained from all guardians at the time of enrollment. Exclusion criteria included a confirmed diagnosis of inborn metabolic disorder (pyruvate carboxylase deficiency, primary carnitine deficiency, fatty acid oxidation deficiency, ketolysis deficiency, or familial hypercholesterolemia), porphyria, impaired liver function, impaired renal function, or cardiovascular disease.

### 2.2. Study Design

The study subjects who received KD were admitted to the pediatric general ward at Kaohsiung Chang Gung Memorial Hospital to undergo a 5-day diet program. All study subjects first underwent metabolic screening to exclude medical conditions that were counter indicative of KD, including blood ammonia and lactate levels, serum cholesterol and triglyceride levels, urinary organic acids, and blood spot tests for amino acid profiles. KD was initiated without fasting during the 5-day hospitalization period. A registered dietitian calculated energy requirements based on daily activity, dietary history, age, body weight, and height. During the 5-day admission period, the patients received one-ninth of the calories on day 1, one-sixth on day 2, one-third on day 3, two-thirds on day 4, and full caloric intake on day 5. The keto ratio started at 2:1 on day 1 and gradually increased to 3–4: 1 on day 5. Daily protein intake was in the range of 1.5–2.5 g/kg/day. Medium-chain triglyceride powder was prescribed at a minimum dosage of 40 g/day for patients with a tolerance for it. Blood sugar was checked every 2–4 h throughout the hospitalization period to prevent hypoglycemia, and β-hydroxybutyrate (βHB) levels were monitored every day to prevent hypoglycemia and hyperketosis. Blood βHB levels were obtained from a fingertip blood sample (Free Style Optium Neo Blood Glucose and Ketone Monitoring System; Abbott Diabetes Care Inc., Whitney, UK) obtained starting on the 2nd day of the 5-day diet program. On the day of discharge, the KD nursery team instructed the families on how to measure urinary ketone levels using a dipstick (Ketostix^®^, Bayer Diabetes, Berkshire, UK). Seizure frequency and βHB levels were recorded using seizure diaries at monthly follow-up visits for 12 months. EEG analysis was performed before initiating the ketogenic diet and at 12 months after the ketogenic diet. We also compared EEG functional connectivity between the control group and study group based on phase-locking value.

### 2.3. Children Behavior Checklist (CBCL)

The child behavior checklist (CBCL) is widely used to detect behavioral and emotional problems in children and adolescents following an intervention [24,25,26,27]. It is a component of the Achenbach System of Empirically Based Assessment [28]. Note that the CBCL is completed by parents using 2 questionnaires for 2 respective age groups: 1.5–5 years and 6–18 years. All patients in the current study exceeded 6 years of age; therefore, we used the CBCL/6–18. This scale comprises eight syndrome scales: anxious/depressed, withdrawn/depressed, somatic complaints, social problems, thought problems, attention problems, rule-breaking behavior, and aggressive behavior. It also includes six DSM-oriented scales consistent with DSM diagnostic categories: affective problems, anxiety problem, somatic problems, attention deficit/hyperactivity disorder (ADHD), oppositional defiant problems, and conduct problems. A total of 113 questions were answered by the parents.

### 2.4. Electroencephalography (EEG) Recording

A total of 37 subjects were recruited for this study. EEG recordings were acquired during the sleep cycle at a sampling rate of 125 Hz. A total of 19 gold-coated silver electrodes (Neuroscan Inc., Sterling, VA, USA) were applied to the skin using conductive paste in accordance with the standard positions of the 10/20 system. Data were recorded continuously on a 32-channel EEG machine (Natus Nicolet One vEEG, San Carlos, CA, USA) at a sampling rate of 2KHz. Several eyes-open and eyes-closed EEG sequences were recorded from from the study group and control group in order to ensure proper spontaneous EEG values, such as eye movement artifacts in the frontal leads and the apparition of posterior alpha rhythm initiated by closing the eyes. EEG impedance values were kept below10 kΩ. To all electrodes was applied a low cutoff filter of 1 Hz, and a high cutoff filter of 70 Hz.

### 2.5. Electroencephalography (EEG) Preprocessing

Continuous scalp EEG data were imported into EEGLAB v2019.0, a MATLAB-based open toolbox [29]. A one-min segment recorded during sleep stage I-II was selected. A finite impulse response filter (0.5–30 Hz) was used to band-pass filter (third-order Butterworth, zero-phase digital filtering) each EEG channel. Data was re-referenced to the average of all scalp channels, and blinks, eye movements, while other mechanical artifacts were removed via independent component analysis (ICA) [30]. Average two independent components (ICs) were removed. The EEG data were subsequently partitioned into 29 epochs with a duration of 4 s and an overlap of 2 s. A thorough examination of the epochs guaranteed that none involved bad channels or contained head motions or muscle movements. Functional networks across broad brain regions were identified based on functional connection strength, which was estimated using the phase locking value (PLV) across all electrodes (Fp1, Fp2, F3, F4, C3, C4, P3, P4, O1, O2, F7, F8, T3, T4, T5, T6, Fz, Cz, and Pz) in the delta, theta, alpha, and beta frequency bands (0.5–4, 4–8, 8–13, and 13–30 Hz, respectively).

### 2.6. Functional Connectivity

PLV was used to independently reconstruct a functional network between all pairs of 19 electrodes for each region of the frequency band and epoch. To moderate the effects of volume conduction, the PLV measure was used to quantify phase synchronization [31]. Note that PLV measures the degree of asymmetry in a distribution of instantaneous phase differences [31]. A symmetric distribution that is centered around zero indicates the possibility of spurious connectivity, whereas a flat distribution indicates a lack of connectivity. Deviations from a symmetric distribution are an indication of dependency between sources. In the current study, the PLV ranged between 0 (no coupling) and 1 (perfect phase locking). Weighted adjacency matrices were produced for each connectivity measure by respectively taking the mean of all 29 functional connectivity matrices for each patient and each frequency band.

### 2.7. Graph Theoretical Analysis

Weighted connectivity matrices were obtained by applying a series of thresholds to the 19 × 19 weighted adjacency matrix of each subject and each frequency band. We used proportional thresholds for network measurements with step-wise increasing values. Despite variations in their values related to graph theory parameters, our results presented a consistent trend [32]. The thresholds were set to 90th, 85th, 80th, …, and 10th percentiles of the weights in the matrix, which resulted in 17 weighted adjacency matrices with densities of 10%, 15%, …, and 90%, respectively. Indexes based on graph theory were used to analyze the weighted connectivity matrices [33]. In graph theoretic analysis, the brain is modeled as a graph composed of nodes and undirected edges, respectively indicating EEG channels and functional connectivity by the PLV. Graph-based parameters were estimated using MATLAB functions collected using the Brain Connectivity Toolbox (www.brain-connectivity-toolbox.net, accessed on 1 September 2021). The following indices were estimated for each of the reconstructed graphs [34]: (1) basic measures (nodal strength), (2) measures of integration (global efficiency), (3) measures of segregation (clustering coefficient), and (4) measures of centrality (betweenness centrality).

### 2.8. Statistical Analysis

We compared the demographic data of patients in the two groups using the Mann–Whitney U test (i.e., age, number, and dosage of Anti-seizure medications (ASMs) and CBCL) and Chi-squared test (i.e., gender, seizure types, etiology of epilepsy, EEG lateralization, and dominant hand). Two-sided Wilcoxon rank-sum tests were used to compare graphs of theoretic FOS properties from patients before and at 12 months after KDT versus the age-matched control group. Comparing network properties before and at 12 months after KDT required that we use the two-sided Wilcoxon signed-rank test to assess cases of FOS presenting right or bilateral EEG lateralization or cases of FOS presenting left or bilateral EEG lateralization. Statistical significance was set at *p* < 0.05, which was corrected for multiple comparisons using the false discovery rate (FDR) [35]. We also performed rank partial correlation coefficient analysis between graph indices, change rates, and CBCL change rates. The CBCL change rate throughout the 12 months assessment period was determined as follows:(baseline CBCL-CBCL in 12 months)/baseline CBCL × 100%.

### 2.9. Study Assessment

#### 2.9.1. Primary Outcome

The number of seizures was recorded by the patient’s family, and the primary outcome was the degree of seizure reduction at 12 months after KD. The formula used to calculate the degree of seizure reduction was [(a – b)/a] × 100, where ‘a’ indicates the number of seizures in the 28-day baseline period and ‘b’ indicates the number of seizures in the 28 days at the 12-month follow-up [36]. Patients with a reduced seizure frequency of ≥50% were defined as responsive and those with a reduced seizure frequency of <50% were defined as non-responsive [36].

#### 2.9.2. Secondary Outcome

After KD treatment, analysis was performed to determine the correlations between changes in EEG functional connectivity and changes in behavioral patterns and emotional state.

## 3. Results

### 3.1. Patient Enrollment

Thirty-seven children were enrolled, including seventeen children with focal onset epilepsy undergoing KDT (study group) and twenty children age-matched control subjects who visited the clinic for headache without intracranial complications nor other medical issues. The mean age of subjects in the study group was 12.03 ± 6.98 years, and the mean age of subjects in the control group was 12.12 ± 3.08 years. No significant difference was observed in the mean age between groups using the Mann–Whitney test (*p* = 0.85). CBCL scores were obtained from parents following the completion of the 12-month KDT program. Note that only 23 of the CBCL questionnaires were filled out (10 fathers and 13 mothers; Appendix A).

### 3.2. Patient Demographics

Table 1 lists demographic data of the study subjects. The mean age in the study group at KD initiation was 12 years. At the time of recruitment, the patients were receiving a median of 2 types of ASM. Among them, 14 (14/17, 82.4%) were responsive after 6 months of KDT, 12 of whom (85.7%) achieved seizure freedom. Note that only 11 of the 14 responsive patients completed the 12-month KD program. Among those who completed the program, 12 (92.3%) saw the frequency of seizures decrease by more than 50% during the 12-month follow-up. One of the patients (No. 9 in Table 1) became responsive during the period between 6 and 12 months after KDT treatment. EEG lateralization included 35% (6/17) on the right side, 29.4% on the left side (5/17), and 35% presented bilateral side with difficulty to determine the origin. Fifteen of the study subjects were right-handed; however, this was not correlated with EEG lateralization patterns. The leading etiology was unknown, whereas the second etiology was genetic factors (data not shown).

### 3.3. Behavioral Assessment

CBCL data are presented in Appendix A. Separate CBCL questionnaires were filled out respectively by the father and mother; however, only ten fathers and thirteen mothers completed the questionnaires, due presumably to familial difficulties. We did not observe a significant difference between the scores assigned by fathers and those assigned by mothers for the syndrome scale or DSM-orientated scales at 12 months after KDT. Nonetheless, the reports by mothers indicated a significant decrease in somatic complaint scores and depressive problems on over the 12 months of KDT program. Overall, it appears that the mothers paid more attention to somatic and emotional complaints than did the fathers.

### 3.4. Functional Connectivity as Compared to Normal Population

In graph theory pertaining to brain functional connectivity, the term ‘node strength’ refers to the sum of network weights attached to ties associated with a given node. A clustering coefficient is used to measure the degree to which nodes cluster together. Global efficiency refers to the ability of the brain to integrate information. Betweenness centrality is a measure of centrality in a graph based on shortest paths; i.e., the level at which the node affects the network. In the current study, network parameters were used to elucidate functional connectivity in patients with epilepsy and an age-matched normal population.

At the time of KD initiation, global efficiency, the mean clustering coefficient, and mean nodal strength of the delta and beta frequency bands were far higher in patients with FOS presenting right or bilateral EEG lateralization than in the controls. This group also presented significantly lower betweenness centrality in the delta and theta frequency bands (Figure 1). Note that in this group, the global efficiency, mean clustering coefficient, and mean nodal strength of the delta and theta frequency bands approached normal values at 12 months after KDT (Figure 2). Nonetheless, global efficiency, the mean clustering coefficient, and mean nodal strength in the beta frequency band were unaffected by KDT. We also observed significantly lower betweenness centrality in the delta and beta frequency bands.

During the study period, FOS patients with left or bilateral EEG lateralization did not present significant changes across proportional thresholds in global efficiency, mean clustering coefficient, mean nodal strength, or mean betweenness centrality in the delta, theta, alpha, or beta frequency bands (Appendix A).

### 3.5. Functional Connectivity after Ketogenic Diet Therapy

Over the 1-year treatment period, FOS patients with right or bilateral EEG lateralization presented a significant decrease in global efficiency in the delta and theta frequency bands, as well as a decrease in mean clustering coefficient and mean nodal strength in the theta frequency bands (Figure 3). No significant changes in global efficiency, mean clustering coefficient, mean nodal strength, or mean betweenness centrality were observed across proportional thresholds in the delta, theta, alpha, or beta frequency bands of FOS patients with left or bilateral EEG lateralization (Appendix A).

### 3.6. Correlation with Children Behavior Checklist (CBCL)

Based on the CBCL/6-18 submitted by fathers, changes in the mean clustering coefficient in the delta band were significantly positively correlated with the change rates of social problems (R = 0.76, adjusted *p* = 0.04) and conduct problems (R = 0.80, adjusted *p* < 0.05) (Figure 4A,B), whereas changes in mean nodal strength in the delta band were significantly positively correlated with the change rates of social problems (R = 0.72, adjusted *p* = 0.03) and conduct problem (R = 0.73, adjusted *p* < 0.05) (Figure 4C,D). Changes in global efficiency were slightly positively correlated with the change rates of social problems (R = 0.75, adjusted *p* = 0.06) in the delta band and with the change rate of thought problems (R = 0.66, adjusted *p* = 0.08) in the beta band. Based on the CBCL/6-18 submitted by mothers, changes in the mean attention deficit/hyperactivity disorder were slightly positively correlated with changes in the mean clustering coefficient (R = 0.64, adjusted *p* = 0.06) and significantly positively correlated with changes in mean nodal strength (R = 0.67, adjusted *p* = 0.03) in the delta band (Figure 4E).

## 4. Discussion

In this study, we determined that KDT was effective in reducing the incidence of seizures and had a noticeable effect on 82.4% of study subjects. Compared to baseline values, patients with FOS with right or bilateral EEG lateralization presented a significantly decrease in the global efficiency of functional connectivity at 12 months after KDT. This indicates that KDT decreased neuronal synchronicity, and in so doing decreased the frequency of seizures. KDT was also shown to moderate behavior problems, by decreasing somatic complaints, depression, and attention deficiency/hyperactivity disorders.

In graph theory, global efficiency indicates the efficiency with which a network transmits information, i.e., integration. The clustering coefficient indicates the degree of connectivity among the neighbors of a given node, i.e., segregation in a local region. Nodal strength indicates the degree to which a given node is able to deliver information. In the current study, we calculated the clustering coefficient and nodal strength of whole brain EEG 19 electrodes and then calculated the average (mean) value. Thus, the mean clustering coefficient and mean nodal strength may indicate the segregation of global regions. We determined that in patients presenting FOS with right or bilateral EEG lateralization, KDT significantly decreased global efficiency, mean clustering coefficient, and mean nodal strength in the theta band. In the delta band, we observed a significant decrease in global efficiency, but no significant difference in the mean clustering coefficient or mean nodal strength, both of which presented curves with *p* values close to 0.05. The above findings appear to indicate that KDT decreased integration and segregation in patients presenting FOS with right or bilateral EEG lateralization. This is similar to the findings in the previous study [22]. In patients presenting FOS with right or bilateral EEG lateralization, we observed no significant differences between the study group and the age-matched control group in terms of global efficiency, mean clustering efficiency, or mean nodal strength. These results indicate that KDT effectively normalized global efficiency (integration), while assisting in seizure control. The above findings suggest that global efficiency is inversely proportional to the seizure threshold. Individuals with higher global efficiency are better able to deliver information, including epileptic discharge, which would make them more susceptible to frequent seizures resulting from a low seizure threshold. From this, it is also possible to infer that if an epilepsy patient initially had higher global efficiency, then KD would have little effect in terms of seizure control. 

Note, however, that patients presenting FOS with left or bilateral EEG lateralization did not present the same results. This prompted our hypothesis that KDT decreased neuronal connectivity in the nondominant brain hemisphere but not in the dominant hemisphere. This finding is worthy of further investigation in the future. We posit that patients with FOS are more likely to benefit from KDT (in terms of seizure control) if their EEG lateralization is toward the nondominant brain hemisphere.

We observed a persistently significant difference in global efficiency in the beta band, even at 12 months after KDT. We posit that this was related to the use of chloral hydrate, Estazolam, or other benzodiazepines during EEG, as indicated in a previous review paper [37]. Under these conditions, it would be easy to misidentify beta activity in graph theoretic analysis. It is also likely that interference by medications also masked the effects of KDT on global efficiency.

Changes in CBCL scores are associated with changes in EEG functional connectivity in specific frequency bands. Overall, the change rate in CBCL scores from specific parents was proportional to the change rate related to the efficiency of EEG functional connectivity. These results suggest that the observed reduction in functional connectivity after 12 months after KDT was associated with the change rate of CBCL scores.

The CBCL questionnaire results from mothers revealed that KDT had significant effects on somatic complaints, depression, and to a lesser degree aggressive behavior, and attention deficit/hyperactivity disorder (ADHD). Note that these findings are in line with previous studies [16,38]. Overall, KDT was shown to have beneficial effects on neuropsychiatric disorders, such as ADHD and depression problem in children.

Functional MRI (fMRI) and/or graph theory are increasingly being used to assess functional connectivity and its correlation with neurological and psychiatric disorders [39,40,41]. In the current study, we used graph theory to investigate KDT-induced changes in EEG functional connectivity and the correlation with changes in CBCL values indicative of behavior changes. Some of the variations in the CBCL questionnaires results were uncorrelated with variations derived using graph theory; however, our results hint at a possible pathophysiology of neuropsychiatric disorders. Further analysis combining fMRI and EEG will be required to fully elucidate the effects of KDT and the underlying mechanisms.

Note that this study was subject to a number of limitations. First, some of the patients suffered from frequent intractable seizures, which made it difficult to filter out epileptiform discharges from the interictal EEG. Second, it was difficult to clearly define the sleep stages of subjects with severe cortical dysfunction. Third, our study cohort was small, due to the fact that focal intractable epilepsy accounts for only a small portion of pediatric epilepsy cases. Fourth, the CBCL is a subjective semi-quantitative scale. We opted not to use an IQ test due to the wide age range in our study group. Fifth, we opted not to analyze the dominant cerebral hemisphere, due to the difficulties that this can impose on children with intellectual disabilities and/or attention deficit hyperactivity disorder. Sixth, we used 19 electrodes EEG and analyzed functional connectivity in terms of phase-locking value. Reducing the number of electrodes to >64 electrodes no doubt reduced low spatial resolution. Future studies will require a high-density EEG set up (>64 electrodes) with functional connectivity estimated after Electrical Source Imaging. Moreover, we did not estimate directionality in the relationship between brain areas in this study which may be related to seizure semiology and treatment effects. We are hoping to include this analysis in our future research.

## 5. Conclusions

This study provided evidence indicating that KDT might be beneficial for pediatric patients with FOS. Our graph theoretic analysis of EEG connectivity suggests that the observed effects were due to a decrease in global efficiency. Note that our findings related to brain connectivity revealed lateralization to the right (non-dominant) hemisphere. Overall, it appears that KDT had positive effects on brain connectivity as well as behavior and emotional states.

## Figures and Tables

**Figure 1 nutrients-14-04457-f001:**
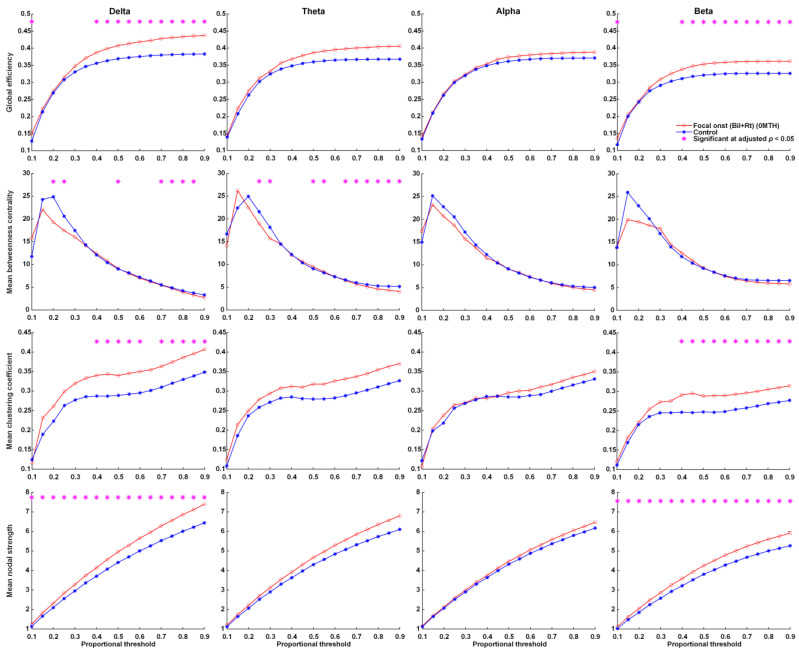
Parameters related to graph theory-based electroencephalographic connectivity in patients with FOS presenting right or bilateral EEG lateralization at initiation and age-matched controls. Asterisks denote statistically significant differences (* adjusted *p* < 0.05).

**Figure 2 nutrients-14-04457-f002:**
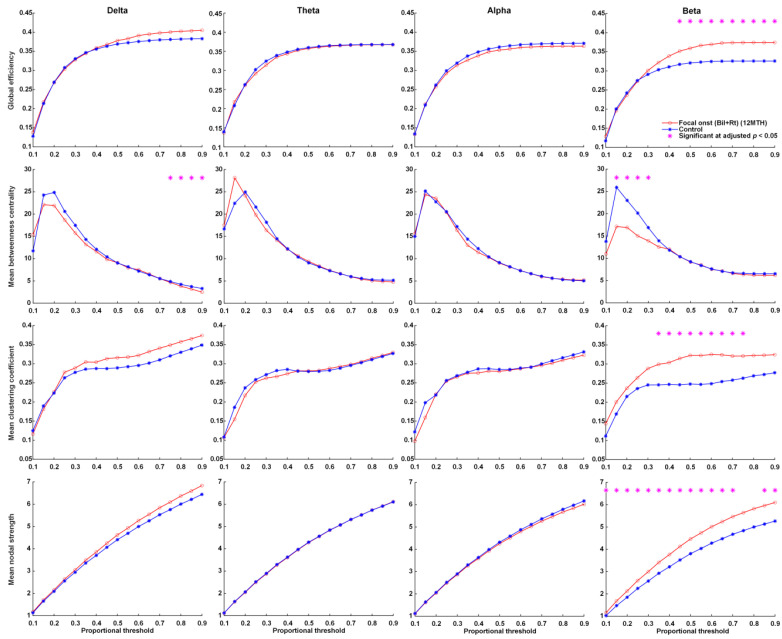
Parameters related to graph theory-based electroencephalographic connectivity in patients presenting FOS with right or bilateral EEG lateralization after 12 months of KDT and age-matched controls. Asterisks denote statistically significant differences (* adjusted *p* < 0.05).

**Figure 3 nutrients-14-04457-f003:**
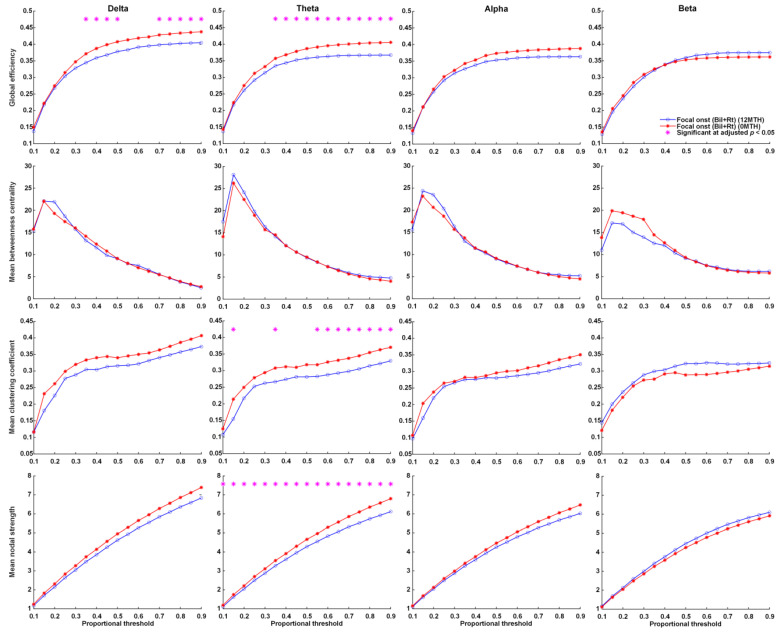
Parameters related to graph theory-based electroencephalographic connectivity in patients presenting FOS with right or bilateral EEG lateralization at KD initiation and after 12 months of KDT. Asterisks denote statistically significant differences (* adjusted *p* < 0.05).

**Figure 4 nutrients-14-04457-f004:**
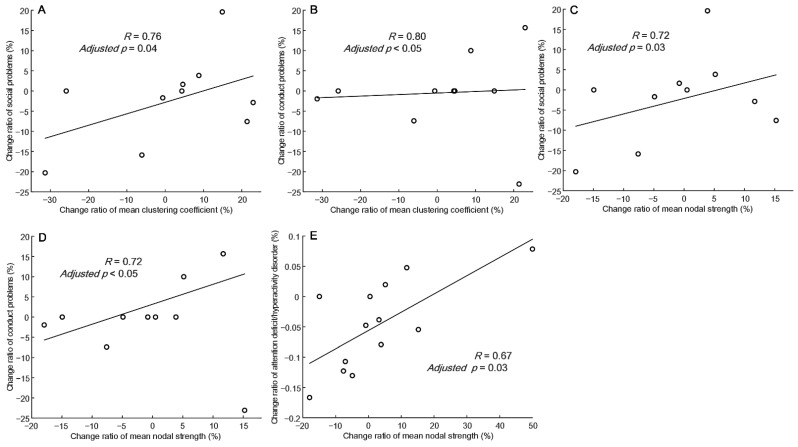
Correlation between the change rate of the mean clustering coefficient and the mean nodal strength in the delta frequency band versus the change rate of CBCL scores from the father (**A**–**D**) and mother (**E**).

**Table 1 nutrients-14-04457-t001:** Demographic data of study subjects.

No.	Age (Year; Months)	Withdrawal (months)	Withdrawn Reason	ASM No.	Baseline Seizure Frequency (fits/m)	Seizure Frequency at 6 m (fits/m)	Seizure Reduction Rate at 6 m (%)	Seizure Frequency at 12 m (fits/m)	Seizure Reduction Rate at 12 m (%)	Responsive (Y/N)	Dominant Hand	EEG Lateralization
1	13; 6	6–9	Parent factor	1	1	0	100.0	-	-	Y	R	L
2	19; 1			2	1	2	−100.0	1	0.0	N	R	R
3	14; 4			3	33	14	57.6	10	69.7	Y	R	R
4	15; 4			4	1	0	100.0	0	100.0	Y	R	Bil
5	6; 10	6–9	Side effect	2	15	0	100.0	-	-	Y	R	Bil
6	10; 3			3	224	0	100.0	0	100.0	Y	L	Bil
7	17; 2			1	1	0	100.0	0	100.0	Y	R	R
8	9; 7			3	1	0	100.0	0	100.0	Y	R	R
9	10; 4			3	66	47	28.8	14	78.8	N	L	Bil
10	0; 10			4	3	0	100.0	0	100.0	Y	R	Bil
11	6; 0	6–9	No effect	1	5	14	−180.0	-	-	N	R	L
12	15; 1			2	1	0	100.0	0	100.0	Y	R	L
13	13; 10			2	6	0	100.0	0	100.0	Y	R	L
14	11; 6			1	1	0	100.0	0	100.0	Y	R	R
15	30; 10			1	2	1	50.0	0	100.0	Y	R	L
16	1; 10			2	56	0	100.0	0	100.0	Y	R	Bil
17	8; 3	6–9	Patientfactor	3	6	0	100.0	-	-	Y	R	R

ASM: anti-seizure medication; F: female, M: male; Responsive, Y: yes, N: no; Dominant hand, R: right hand, L: left hand; EEG lateralization, R: right side, L: left side, Bil: bilateral side.

## Data Availability

The datasets generated and/or analyzed during the current study are not publicly available due to a licensing agreement.

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
