# Peer review of "Graph Theory-Based Electroencephalographic Connectivity via Phase-Locking Value and Its Association with Ketogenic Diet Responsiveness in Patients with Focal Onset Seizures"

_nutrients, 2022, doi:10.3390/nu14214457_

Round 1

Reviewer 1 Report

Thank you for inviting me to review this manuscript. This study is essential regarding epilepsy and the ketogenic diet and its association with the cortical network. 

INTRODUCTION

The author described the background of this study thoroughly. 

METHOD

What was the reason for choosing children with headaches as a control? 

Line 166: “Several eyes-open and eyes-closed EEG sequences were recorded 166 from the volunteers in order to ensure proper spontaneous EEG values.. “ What were volunteers referred to? Were they normal volunteers or epilepsy patients? It would be better if the author use the same terminology for subjects along the manuscript (i.e. control and FOS epilepsy patient)

Line 173: “A one-min segment recorded during sleep stage I-II was selected.” Was it normal sleep or induced sleep with sedation? Sleep stage with sedation usually was prominent in sleep stage III-IV rather than sleep stage I-II. 

The abbreviation of AED (Line 227) should be written before it was abbreviated. However, the term Anti-seizure Medication (ASM) is more frequently used nowadays. 

RESULT

Line 313: there is a description of Figure 1. But there wasn’t any figure above this description 

DISCUSSION

The author discussed the result thoroughly and explained the association between the ketogenic diet and cerebral network findings. 

Line 391: “We posit that this was related to chloral-hydrate, Estazolam, or other benzodiazepines during EEG….”. Did all subjects have sleep induced by drugs? 

CONCLUSION

The author made the conclusion based on the findings. 

Reviewer 2 Report

KDT is vital treatment of refractable epilepsy, particulayrly for children. The authors demostrated a study by showing the assotiation between EEG connectivity and treatment outcome. However, they are a few points needed to be clarified:1) Since all the enrolled patients FOS, are they treated already by ASMs? how many types ASMs, are they underwent MDT evaluation for surgery? or some of them are treated by VNS or other surgeries. ALL these info are missing. 2)The authors have publised "Su TY, Hung PL, Chen C, Lin YJ, Peng SJ. Graph Theory-Based Electroencephalographic Connectivity and Its Association with Ketogenic Diet Effectiveness in Epileptic Children. Nutrients. 2021;13(7):2186. Published 2021 Jun 25. doi:10.3390/nu13072186" I did not see too much novelty for the current study.

Round 2

Reviewer 2 Report

The authors have answered my concerns properly. however, as a neurosurgeon, I believe if the lesion of the patient is respectable, surgery is the best choice for Focal epilepsy patients. KDT is still difficult for the children to last for long period.